# Cell-in-Cell Structures in Gastrointestinal Tumors: Biological Relevance and Clinical Applications

**DOI:** 10.3390/jpm13071149

**Published:** 2023-07-17

**Authors:** Irina Druzhkova, Nadezhda Ignatova, Marina Shirmanova

**Affiliations:** Research Institute of Experimental Oncology and Biomedical Technologies, Privolzhsky Research Medical University, 603005 Nizhny Novgorod, Russia; ignatova_n@pimunn.net (N.I.); shirmanova_m@pimunn.net (M.S.)

**Keywords:** cell-in-cell, CIC, cancer, entosis, cell engulfment, cell internalization, cancer prognosis, drug resistance

## Abstract

This review summarizes information about cell-in-cell (CIC) structures with a focus on gastrointestinal tumors. The phenomenon when one cell lives in another one has attracted an attention of researchers over the past decades. We briefly discuss types of CIC structures and mechanisms of its formation, as well as the biological basis and consequences of the cell-engulfing process. Numerous clinico-histopathological studies demonstrate the significance of these structures as prognostic factors, mainly correlated with negative prognosis. The presence of CIC structures has been identified in all gastrointestinal tumors. However, the majority of studies concern pancreatic cancer. In this field, in addition to the assessment of the prognostic markers, the attempts to manipulate the ability of cells to form CISs have been done in order to stimulate the death of the inner cell. Number of CIC structures also correlates with genetic features for some gastrointestinal tu-mors. The role of CIC structures in the responses of tumors to therapies, both chemotherapy and immunotherapy, seems to be the most poorly studied. However, there is some evidence of involvement of CIC structures in treatment failure. Here, we summarized the current literature on CIC structures in cancer with a focus on gastrointestinal tumors and specified future perspectives for investigation.

## 1. Introduction

Cell-in-cell (CIC) formation is a phenomenon where one cell engulfs another one with or without its death [1,2]. The formation of CICs is usually described by many different terms, such as entosis, engulfment, emperipolesis, and cannibalism [3]. CIC structures were first discovered in tissue samples around 120 years ago and initially called ‘bird’s eye’ structures by the investigators [4]. The first description of a CIC was originally made in the second half of the 19th century, when Karl J. Eberth reported lymphocytes inside intestinal epithelial cells [5]. In normal physiology, for instance, during early pregnancy, uterine epithelial cells are internalized by blastocyst trophoblast cells to facilitate embryonic implantation. Similar cell engulfment events can also occur between pairs of blastomere cells. Then, such structures were discovered for homotypic white blood cells, and the mechanism of the active penetration of one cell into another was described [6,7]. They noticed that CIC formation was typical both for cells of the immune system and for intestinal cells.

CIC structures were found in cancers and proposed as diagnostic markers of malignancy at the beginning of the 21st century [8]. Furthermore, it has only been during the recent decades that such a phenomena have become the subject of an active research, both on cell cultures and pathomorphological materials. This has become possible mainly because of the fast development of appropriate microscopic techniques such as scanning electron microscopy, confocal fluorescence microscopy, super-resolution fluorescence microscopy, etc. In addition, the use of exogenous fluorescent dyes and genetically encoded fluorescent proteins provides improved opportunities to visualize the CIC formation process and investigate agents involved in its modulation. Genetic technologies allow to search for genetic markers that are specific for cells inclined to CIC formation, as these might serve as diagnostic criteria or as targets for therapeutic impact. 

Most studies have demonstrated that CIC structures have a pro-tumorigenic role in different cancer types [9,10,11,12,13,14,15,16]. Typically, the majority of inner cells undergo degradation by autophagy within the outer cells, as with cell death mechanisms. This confers a survival advantage of the outer cells under stress conditions. However, in some cases, the inner cell can undergo cell division within the outer cell or even escape. Overall, the functions and the development of CIC structures in cancer remain poorly understood.

Meanwhile, understanding their fundamental role within various types of malignancies and determining the correlation with patient outcomes can help identify novel prognostic markers.

## 2. CICs in Cancer: Types, Terms, and Biological Impact

CIC structures in cancer tissues can be divided into two major groups: homotypic and heterotypic. A homotypic CIC is formed when one epithelial cell engulfs another one. A heterotypic CIC results from the fusion of an epithelial cell and a cell of a different nature, normally an immune system cell [17]. The functions and outcomes of such structures can differ depending on the particular situation (Figure 1). Sometimes the engulfed cells can undergo cell division inside the outer cell, or they may escape. However, the majority of engulfed cells undergo degradation inside the outer cells by an autophagic, lipidation-dependent, lysosomal cell death mechanism known as entotic cell death [18]. Notably, under stress conditions, entotic cell death can confer a survival advantage for the outer cells [19,20].

In the literature, the phenomenon of the formation of cell-in-cell structures is described by several basic terms, such as cannibalism, emperipolesis, and entosis.

Cannibalism (from canı’bal, in Spanish, in reference to alleged cannibalism among Caribs) is the phenomenon in which one cell surrounds and engulfs another one in order to digest it [21]. It was the first term to describe cell-in-cell structures. Such cell structures are characterized by a crescent-moon-shaped nucleus of the host cell as the nucleus is displaced to the periphery by a large vacuole containing ingested cells [22]. This phenomenon is considered to be one of the signs of malignancy [23,24,25].

Cannibalism can be observed both between cells of the same origin and between cells of different origin. For example, cannibalism of neutrophilic tumor cells was observed in 1.4% of cases in a study of 500 cases of oral squamous cell carcinoma. [26]. There is another form of cannibalism, in which one malignant cell engulfs another, and this complex is then engulfed by another cell. In addition, one cell can absorb two cells at once. This phenomenon is called “complex cannibalism” [27].

Cannibalism can serve as a sign of tumor progression since, in a number of studies, for example, in melanoma, this phenomenon is observed only in metastatic cells, and not in cells of a primary tumor. Cannibalism can also be a mechanism for tumor escape from immunity by “eating” immune cells [28,29].

The process of cannibalism is associated with phagocytosis, a common cellular phenomenon that plays an important role in inflammation resolution, antigen presentation, and removal of apoptotic cells [30].

The cells responsible for phagocytosis can be divided into two main types: professional phagocytes, such as macrophages and neutrophils, and lay phagocytes, such as epithelial cells [31].

However, there are important differences between cannibalism and phagocytosis. First, the goal of cannibalism is to provide the host cell with nutrients [23,32], while the goal of phagocytosis is mainly to eliminate the infectious agent [30]. Second, cannibal cells prefer to engulf living related cells, while cells of the immune system phagocytize dead cells [33]. Thus, Lugini et al. demonstrated that co-culturing of metastatic melanoma cells with live lymphocytes resulted in a high rate of cannibalism of immune cells by cancer cells, allowing the metastatic cells to survive even under serum-deficient conditions. On the contrary, the use of latex beads did not promote survival, and cancer cells quickly died, which indicates a complex system of communication between cells that regulates the process of cannibalism [23].

Emperipolesis comes from the Greek (em—within; peri—around; polemai—to roam). It was first described 50 years ago as the active penetration of one cell into another, which remains intact [7]. It has been suggested that the terms cell-in-cell and “emperipolesis” can be used as generic terms for cell-in-cell structures or related cell movements, while entosis, cannibalism, and cytophagocytosis should be used to discuss the specific mechanisms of cell-in-cell formation [1].

Emperipolesis includes heterotypic cell-in-cell structures that are mainly formed from histiocytes and megakaryocytes, but also occur in tumor tissue [34], such as neutrophil cells engulfed by megakaryocytes in the bone marrow [35] and thymocytes taken up by thymic feeder cells in the thymic cortex [1,36]. Emperipolesis has also been described in renal cell carcinoma [37], squamous cell carcinoma [38], and other cancers, as well as in infectious liver diseases [39]. Natural killer (NK) cells can induce remission of liver fibrosis by killing hepatic stellate cells (HSCs).

It has been shown that NK cells from patients with liver cirrhosis infiltrate HSCs, impairing the antifibrotic capacity of NK cells through TGF-β-dependent emperipolesis [40]. The fate of engulfed cells differs from that of cannibalism, as engulfed cells can escape their host [41] or undergo mitosis [34,42]. Since immune cells and tumor cells compete for survival, the co-survival of these two cells is similar to the phenomenon of commensalism.

The biological significance and mechanisms of emperipolesis require further study [43]. It involves only living cells, while cannibalism can use dead or dying cells and other materials in the extracellular space [44].

The engulfed cell can be destroyed, and different terms will be used depending on how it dies. For example, CD8+ T cells in hepatocytes undergo non-apoptotic death and are thought to actively enter hepatocytes rather than being engulfed by them. This process is called suicidal emperipolesis [45,46]. Granzyme B is an enzyme that exists mainly in NK cells and cytotoxic T lymphocytes and can induce apoptosis [47]. Granzyme B released from the inner NK cell is taken up by vacuoles, which, in turn, induce NK cell apoptosis. This process, called emperitosis (combining emperipolesis and apoptosis), describes how cytotoxic lymphocytes in tumor cells are killed by apoptosis, allowing tumor cells to elude the immune response [48].

The host cell can also be destroyed; destruction of tumor cells containing lymphocytes has been observed [49,50]. Emperipolesis is believed to be the pathway by which NK cell-mediated tumor cell death is regulated [34]. It is also suggested that these structures are involved in modulating the tumor microenvironment using unique mechanisms [43].

Compared to cannibalism and emperipolesis, entosis is a relatively new concept of a cell within a cell that has been proposed in recent years. The name comes from the Greek word “enthos”, which means “within or inside”. First observed in human mammary epithelium, this is a homogeneous living cell phenomenon that occurs in human epithelial and other tumors, including invasion of one living cell into another, followed by degradation of internalized cells by lysosomal enzymes [51].

It is well known that cells undergo apoptosis when they lose contact with the extracellular matrix (ECM) or attach to inappropriate sites, in a process called anoikis [52]. Anoikis prevents tumor formation or metastasis. However, anoikis inhibition is insufficient for long-term survival of tumor cells separated from the extracellular matrix, while entosis may promote the survival of detached cells [51].

Matrix deadhesion seems to induce entosis [53], and indeed, this phenomenon is observed mainly in suspended samples, such as liquid exudate, urine, and bile [34]. However, recent studies have shown that entosis can also occur when cells attach to the matrix [54,55,56]. In this case, mitosis is the stimulating factor.

Entosis is believed to promote competition between tumor cells, providing nutrition to the host cell when the engulfed cells die. This promotes proliferation and metastasis of tumor cells [57].

Entotic cells usually die, but they can also escape the host. A small percentage of cells undergo cell division inside the host cell [51]. The death of entotic cells is carried out in a non-cellular-autonomous manner due to the autophagy pathway-dependent lysosomal degradation of living, engulfed cells [58]. Cell death occurs in the absence of caspase-3 cleavage and without morphological signs of apoptosis. It has been suggested that entotic cell death should be defined as the death of new type IV cells [59].

The fate of entotic cells shows that entosis can inhibit tumor progression by killing tumor cells that have detached from the matrix. However, recent studies show that entosis may also promote tumor progression by inducing changes in cell ploidy [60]. Entosis directly promotes polyploidy by disrupting cell division, resulting in the formation of polyploid cells in culture [61].

Correlations between the presence of CIC structures and the stage of cancer have been demonstrated, for example, in lung, stomach, breast, renal, and pancreatic cancers and in head and neck squamous cell carcinomas [10,11,12,13,14,15,16]. Additionally, a simple comparison of the growth of xenografts in mice revealed the development of larger tumors from cells with higher entotic activity, which indirectly indicated the pro-tumorogenic function of CIC structures [62]. However, in vivo studies on animal models do not provide a clear idea of the involvement of CICs in influencing the growth or metastatic potential of tumors [43,62,63]. 

In early studies, it was suggested that CICs mostly contribute to cell death and therefore have an anti-tumor effect [51,64]. Then, evidence demonstrating the ability of significant proportions of cells to escape from the host cells began to accumulate [3,62]. Now it is generally considered that the host cell provides a safe environment in which the inner cell can avoid unfavorable conditions such as nutrient starvation, toxic agents, or even immune cell attacks [19,55,65,66]. The role of CIC structures in the promotion of clonal selection and tumor evolution has also been shown [13,67]. The other evidence of pro-tumor effects is the formation of heterotypic structures, whereby a cancer cell engulfs an immune cell with the subsequent death of the latter, resulting in failure of the immune response [32,46]. These clinical histopathological studies of CIC structures in human cancers are summarized in Table 1. Collectively, they show that the presence of CICs mainly indicates an unfavorable prognosis due to elevated metastatic activity and cancer cell survival.

The possibility of identifying CIC structures on histopathological slides from patients’ tumors provides a field for the development of new prognostic markers useful for the stratification of cancer patients for appropriate treatment. 

## 3. CICs in Cancer: Mechanisms of Formation

Numerous factors initiating the process of engulfment have been determined, e.g., matrix detachment [55,78], the composition of membrane lipids [79], glucose starvation [19], serum components [80], proinflammatory environment (cytokine, growth factors, and ROS) [81], proinflammation, and mitosis (Figure 2) [55,82]. Two core events, cadherin spanning and actin–myosin contraction, orchestrated by Rho GTPases, have also been described [13]. 

Recently, another core element—a mechanical ring, enriched with vinculin and ezrin and located between adherent junctions and contractile acto-myosin, has been identified [83]. However, the specific stages, participating molecules, and ultimate prevalence of the inner or outer cells depend on the specific type of engulfment. 

Thus, the ‘driver cell’ may be the one invading the external cell, this being more likely for entosis [19,84]; conversely, the driver cell can be one that extends protrusions around the soon-to-be engulfed cell, effectively a case of cannibalism [85,86]. 

For homotypic CIC structure formation the initial contact is normally accomplished not through ligand/receptor interaction, but through the membrane components. In the case of entosis it was demonstrated that cell-in-cell structures cannot form in tumor cells with insufficient E- and P-cadherin, key components of the adherent junctions of cells, but they occur when E- or P-cadherin is re-expressed [20,87,88]. Alpha-catenin participation in cell-in-cell formation was also shown [89]. Actomyosin contraction between neighboring cells is the following step of entosis [44]. This process is mediated by the small GTPase RhoA and its effector kinase rho-kinase (ROCK I/II) [20,51]. Cell-in-cell structure formation is blocked when these are inhibited [90]. As noticed above, mitosis drives entosis in adherent cells. This process is regulated by CDC42, a protein with a role in the attachment of cells to each other and to the ECM. CDC42 has been shown to increase mitotic de-adhesion and rounding. This may be because CDC42 constrains mitotic RhoA activation in polarized epithelial cells, and its loss causes RhoA overactivation during metaphase [55] (Figure 3B).

The molecular mechanism of cannibalism involves caveolins, ezrin, and TM9. It has previously been shown in caveolin (Cav)-1, Cav-2, and Cav-3; additionally, in Cav-1 and Cav-2, major structural proteins of caveolae infectious tumor metastasis [26]. The main crosslinking agent between cortical actin filaments and plasma membranes is ezrin. It regulates cytoskeletal organization by increasing the rate of roguanosine 5’-triphosphatase (GTPase) signaling [91] and is expressed on phagocytic vacuoles of melanoma cells that occur in cannibalism [92] (Figure 3A). Ezrin also promotes communication between actin and caveolin-1-enriched tumor cell vacuoles, which form the driving structure of the cannibalistic process [23]. Influencing this structure can suppress cannibalism [21]. TM9 is a highly conserved protein with nine transmembrane segments. It may play a key role in phagocytosis, adhesion, and nutrient uptake [93].

TM9SF4, a member of the TM9 superfamily (TM9SF) in humans, is overexpressed in metastatic melanoma cells but is not found in cells of primary lesions. Knockdown of TM9SF4 inhibits the phenomenon of cannibalism [94]. Previously, it was shown that increased acidity in the microenvironment can serve as an inducer of cannibalism. In turn, TM9SF4 can bind to the ATP6V1H subunit of the proton pump with active V-ATPase, which regulates the pH gradient in tumor cells [32].

While the mechanisms of formation of cell-in-cell structures in entosis and cannibalism have quite a lot in common with those in emperiopolesis due to the mandatory participation of immune cells, this process has its own characteristics. Lymphocyte function-associated antigen-1 (LFA-1 or CD11a/CD18) can mediate intercellular interactions between leukocytes and non-blood cells. Together with its ligand, intercellular adhesion molecule 1 (ICAM-1/CD54) [95], it is thought to be associated with emperipolesis, which can be blocked by co-administration of an anti-LFA1 antibody in vivo (Figure 3C) [1]. Takeuchi et al. created a novel human Treg cell line, HOZOT, from umbilical cord blood mononuclear cells. These cells are able to actively invade target cancer cells and form cell-in-cell structures classified as emperipolesis. 

MHC class I appears to be the main molecule used by HOZOT cells for target cell recognition, and ICAM-1 is one of the molecules recognized by HOZOT cells. In addition, anti-CD62L causes partial inhibition of the intercellular activity of HOZOT cells [50]. Normal transcellular migratory activity and the receptors that regulate it are considered important for emperipolesis [1]. These include extracellular free calcium and adhesion molecules, as well as the actin-based cytoskeleton and ezrin [34]. Abnormal P-selectin localized in the demarcation membrane system of neutrophils and megakaryocytes is associated with their contraction and has been proposed as a cause of emperipolesis in bone marrow fibrosis [96].

Whatever the mechanism, the results of such interactions will be cell-in-cell structures, which can be detected and, based on the outcome of the process, can serve as prognostic markers [2]. 

## 4. CICs in Gastrointestinal Tumors

The most significant studies of CICs in gastrointestinal cancers of different localization are presented in the Table 2.

### 4.1. Pancreatic Cancer

Among all gastrointestinal tumors, most of the research on CICs has been devoted to pancreatic cancer. For example, Hayashi et al. observed that CICs occurred more frequently in liver metastases than in primary pancreatic ductal adenocarcinoma (PDAC). Greater numbers of CICs correlated with a decrease in the degree of tumor differentiation. With respect to genetic features, TP53 mutations, KRAS amplification, and MYC amplification were associated with the presence of CICs in PDAC tissues [12]. Mlynarczuk-Bialy et al. demonstrated that, in BxPC3 PDAC cell cultures, there is very active CIC formation and an ~80% survival rate of the inner entotic cells [2]. Mackay et al. observed greater numbers of CIC structures in H&E-stained sections from mutant p53 pancreatic tumors than in those from p53 null xenografts. Tripolar mitoses in CIC structures were often observed in mutant p53 mice but not in p53 null mice. Overall, CIC formation is considered a factor contributing to genetic and chromosomal instability [62,97]. 

In a study by Gast et al., circulating heterotypic CICs comprising immune and epithelial cells were found in the peripheral blood of patients with PDAC. The presence of such structures was directly correlated with the stage of the tumor and was inversely correlated with overall survival [98]. In a study by Jang et al., the presence of CICs in the CD44^high^/SLC16A1^high^ pancreatic cancer cell line was able to enhance anchorage-independent growth or invasiveness [99].

Attempts to manipulate the ability of cells to form CICs have been made in a few studies. Methylseleninic acid (MSeA) was shown to induce CIC structure formation in pancreatic cancer cells through entosis regulated by CDC42 and CD29, and this process led to the death of the inner cell [100,101]. The inhibition of nuclear protein 1 (nupr1), a potential mediator of PDAC resistance to cellular stress, led to homotypic cell cannibalism with subsequent cell death [85,102]. Therefore, an association between the presence of CIC structures and good prognoses was demonstrated.

### 4.2. Stomach Cancer

Stomach cancer appears to be the most poorly investigated gastrointestinal type in terms of CIC formation. The possibility of stomach cancer cells forming CICs has been demonstrated in vitro [51]. Barresi et al. demonstrated the phagocytosis of immune cells by highly invasive and aggressive cancer cells of micropapillary carcinomas of the stomach [103]. However, any correlations with disease progression or treatment response are absent as long as the detailed descriptions of mechanisms of CIC formation for that cancer type.

### 4.3. Hepatocellular Cancer

It is known that hepatocytes are “non-professional” phagocytes, and phagocytosis by them can lead to the formation of CIC structures [104]. Several recent studies have reported on the participation of hepatocytes in almost all types of neighboring cell engulfment, such as efferocytosis, and live cell internalization events, including suicidal emperipolesis, entosis, and enclysis [45,105,106]. CIC structures have long been observed by histopathologists in some chronic liver diseases, such as autoimmune hepatitis [107,108] and chronic viral infection [109,110]. The potential role of CICs in liver injury or T-cell clearance has been widely discussed [46,111]. One of the latest studies, by Su et al., described the process of host-cell death as a result of the engulfment of natural killer cells (NKs), the effect being called “in-cell killing”. Moreover, the action of CD44 on tumor cells was identified as being a negative regulator of “in-cell killing” via its inhibition of CIC formation. The antibody-mediated blockade of CD44 signaling potentiated the suppressive effects of NK cells on tumor growth associated with increased heterotypic CIC formation, suggesting a new potential approach for cancer immunotherapy [112].

### 4.4. Colon Cancer

A study by Bozkurt et al. demonstrated the prevalence of homotypic CIC structures at the invasive fronts of colorectal tumors. Moreover, the activation of entosis in colon cancer cells by TRAIL signaling has been experimentally proven, and a marked elevation in TRAIL expression in histopathological slides with CIC structures has additionally been confirmed [113]. Furthermore, a positive correlation between CIC formation and tumor malignancy was observed in a study by Wang et al. In this study, internalization of cytotoxic lymphocytes into cancer cells was demonstrated. The promotion of death of the internalized lymphocytes in the CIC structures by the inflammatory cytokine IL-6 was also revealed, suggesting the contribution of CICs to tumor immune escape [9]. 

**Table 2 jpm-13-01149-t002:** Studies of CICs in gastrointestinal tumors.

Cell Origin	Type of CIC	Model	Indicator	Reference
Esophageal cancer	Tumor cells + macrophage, epithelium-macrophage-leukocyte, homotypic CICs	Patients’ tissue samples, ICC	favorable prognosis	Wang, 2021 [68] Cui, 2021 [69]
Pancreatic ductal adenocarcinoma	Tumor cells + macrophage	orthotopic transplantation into mice (in vivo)	aggressive metastatic spread of cancer	Clawson, 2017 [63]
Colorectal cancer	Tumor cells + macrophage, colorectal cancer + monocytes, homotypic CICs,	cell cultures (in vitro), ICC	metastatic potential, adverse patient prognosis	Bozkurt, 2021 [113] Montalbán-Hernández, 2022 [72]
Hepatocellular cancer	Tumor cells + macrophage, homotypic CICs	In vitro (cell fusion), in vivo (in mice), ICC	resistance to chemotherapy, promoting progression of more aggressive tumors	Wang, 2016 [78] Davies, 2020 [104]
Intestinal epithelial cells	intestinal epithelial cells and macrophages	in vivo (in mice)	metastatic conversion of cancer cells	Powell, 2011 [74]
Gastric micropapillary carcinoma, gastric adenocarcinomas	Neutrophils + tumor cells, mesenchymal stem cells + GIT epithelium	Microscopy of patients’ samples, ICC	aggressive behavior and poor prognosis transform into malignant cells	Barresi, 2015 [103] Caruso, 2002 [114] Houghton, 2004 [75]
Anal cancer	homotypic CICs	Microscopy of patients’ samples, IHC	better survival	Schwegler, 2015 [15]
Rectal cancer	homotypic CICs	Microscopy of patients’ samples, IHC	tumor progression metastatic clear cell cancer	Kong, 2015 [13]

In a study by Schwegler et al., paraffin-embedded samples of patients’ tumors were analyzed, and a correlation between low CIC rates and improved prognosis was shown for anal cancer, while, for rectal cancers, low rates were associated with longer, local failure- and metastasis-free survival [15]. In elegant in vivo research by Powell et al., fusion of cancer cells and macrophages was demonstrated, and this process impacted the physical behavior attributed to migratory macrophages, including their navigation of the circulation, contributing to the metastatic conversion of cancer cells [74].

In our study, the presence of homotypic CICs in histopathological slides of colorectal cancers was shown, and this correlated with the in vitro formation of CIC structures by cancer cells isolated from the corresponding tumors and the longer survival of cancer cells in the culture conditions (Figure 4).

## 5. Conclusions and Perspectives

CIC structures can be identified in almost all types of cancer. Both types of CIC structures, homotypic (tumor cell inside tumor cell) and heterotypic (tumor cell inside immune cell or immune cell inside tumor cell), have been found in various cancers but can result in different fates of the engulfed cells. Although a positive prognostic value of the presence of CICs in tumor tissues was revealed in a few studies, this is rather an exception to the general rule. Most research has demonstrated a positive influence of CICs on tumor progression, with this being shown for different cancers, including gastrointestinal ones. 

Assessment of the role of CIC structures in the responses of tumors to therapies, both chemotherapy and immunotherapy, is especially interesting and clinically relevant. While several studies have shown that stimulation of CIC formation occurs upon drug exposure and as a result of immune cell attacks, the existing data are insufficient to suggest any specific mechanisms, indicating that further studies are urgently needed in this field. 

The high incidence of CICs in gastrointestinal tumors and contradictions in the available data suggest that this, too, is a field requiring fundamental research aimed at discovering the mechanisms of CIC formation and ways to control this process. The possibility of identifying CICs in standard histopathological slides routinely inspected by clinical pathologists means that the presence of CICs represents a simple but promising prognostic marker, but we need a better understanding of what such a presence can lead to in different situations. 

## Figures and Tables

**Figure 1 jpm-13-01149-f001:**
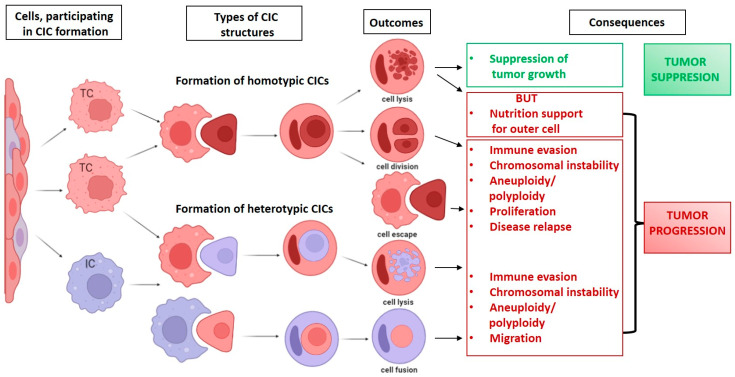
Types of CIC and consequences.

**Figure 2 jpm-13-01149-f002:**
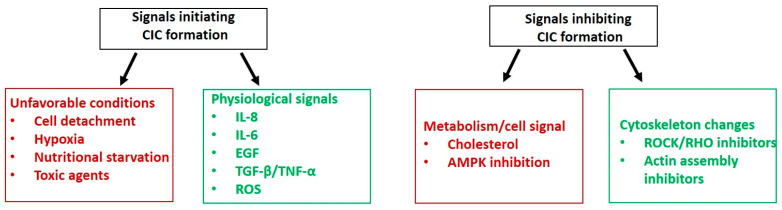
The initiating and inhibiting signals for CIC formation.

**Figure 3 jpm-13-01149-f003:**
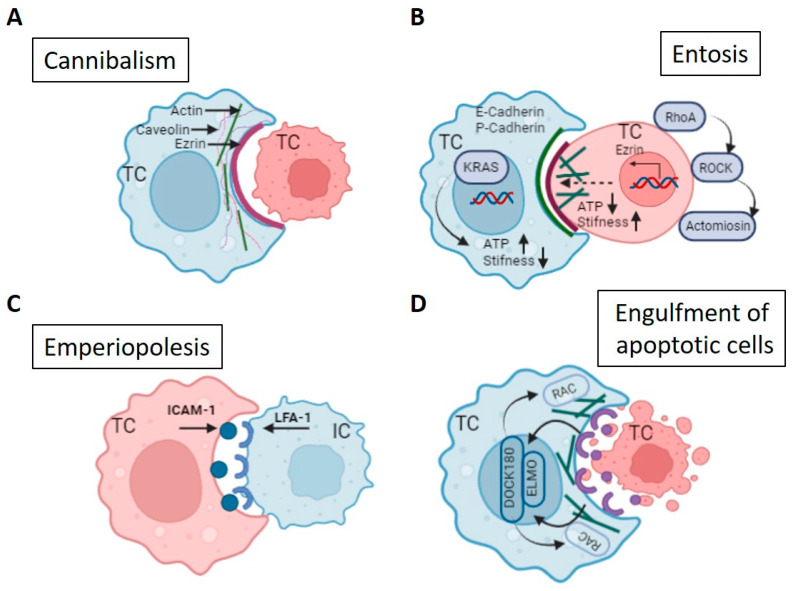
Mechanisms of cell engulfment: (**A**) Tumor cells (TCs) can cannibalize live cells as well, for example, TC. Cortical protein ezrin, vesicles of the caveolar network, and actin filaments are the main participants of this process. (**B**) One TC engulfs another TC in the entotic process, where inner cell (pink) is an active participant, with higher stiffness, which is regulated by Rho-associated protein kinase (ROCK). E-cadherin or P-cadherin provide the initial cell–cell adhesion. Entosis involves transcriptional upregulation of ezrin through myocardin-related transcription factor (MRTF) that is recruited to the nucleus in response to increased cortical tension. KRAS activation and relative high-energy states are linked to outer cell identity owing to a relative reduction in cell stiffness compared with inner cell. (**C**) Formation of heterotypic CIC often associated with ligand/receptor interaction, particularly, ICAM-1/LFA-1 interaction. (**D**) Engulfment of apoptotic cell, also realized through the ligand/receptor interaction in response to “eat-me” signal. The best-characterized signal is phosphatidylserine (PS purple), which is normally exposed on the outer leaflet of apoptotic cells. PS can bind to at least three host-cell receptors, including brain-specific angiogenesis inhibitor-1 (BAI1), stabilin-2, and T-cell immunoglobulin and mucin domain-containing protein-4 (TIM4). Host-receptor activation initiates a signaling cascade that leads to activation of RhoG through the nucleotide-exchange factor (GEF) complex 180 kDa protein of CRK (DOCK180) and cell motility protein (ELMO), leading to activation of small GTPase protein Rac and following cytoskeleton reorganization of outer cell.

**Figure 4 jpm-13-01149-f004:**
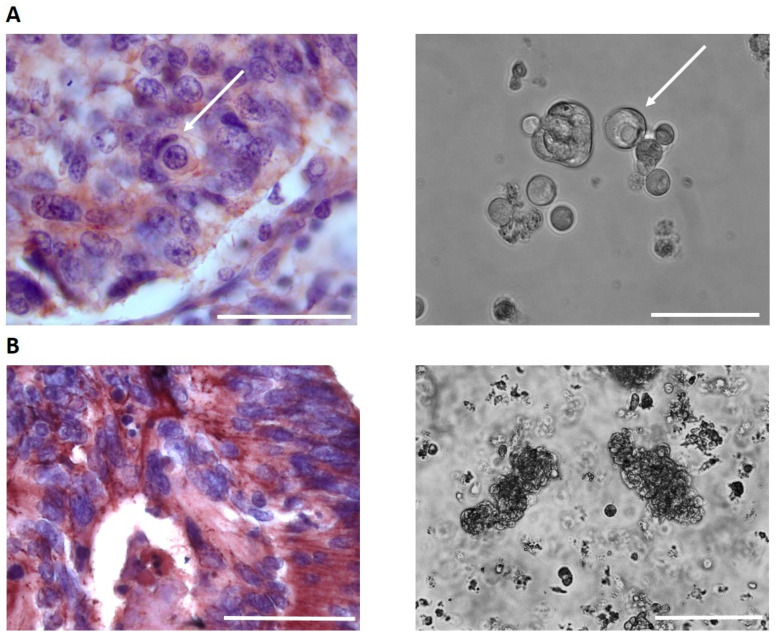
Anti-E-cadherin immunohistochemical staining of colorectal cancer samples and primary cell culture from the colorectal tumor: (**A**) presence of cell-in-cell structures; (**B**) absence of cell-in-cell structures. Arrows indicate CICs. Bar is 100 µm for all images.

**Table 1 jpm-13-01149-t001:** Histopathological studies of CICs in human cancers.

Cancer Type	CIC Type	Function	Reference
Esophageal cancer	Tumor cells + macrophage, epithelium–macrophage–leukocyte, homotypic CICs	Favorable prognosis	Wang, 2021 [68] Cui K, 2021 [69]
Lung cancer	Tumor cells + monocytes	Increased migratory or invasive properties	Aguirre, 2020 [70] Miroshnychenko, 2021 [71]
Melanoma	Tumor cells + NK, tumor cells + T-cells, macrophage–tumor cell fusions	Increased tumor cell survival, dissemination, local recurrences and resistance to immunotherapy	Gutwillig, 2022 [65]
Colorectal cancer	colorectal cancer cells + monocytes	Facilitated metastases and aggressive phenotype	Montalbán-Hernández, 2022 [72]
Pancreatic ductal adenocarcinoma	macrophage + tumor cell fusions	Cancer progression	Nitschke, 2022 [73]
Intestinal epithelial cells	epithelial cells + macrophages	Transformation into malignant cells, increased migratory or invasive properties	Powell, 2011 [74]
Gastric carcinoma	mesenchymal stem cells + GIT epithelium	Transformation into malignant cells	Houghton, 2004 [75]
Breast cancer (somatic cell fusions)	basal-like and luminal breast cancer cells	Metastatic progression and therapeutic resistance	Su, 2015 [76]
Ovarian cancer	hematopoietic cells and epithelial cancer cells	Increase of invasive properties	Ramakrishnan, 2013 [77]
Anal cancer	Homotypic CICs	Improved prognosis	Schwegler, 2015 [15]
Rectal cancer	Homotypic CICs	Poor prognosis	Schwegler, 2015 [15]
Head and neck squamous cell carcinoma	Homotypic CICs	Poor prognosis	Schwegler, 2015 [15]
Renal clear cell cancer	Homotypic CICs	Correlation with high grade of disease and metastasis	Kong, 2015 [13]

## Data Availability

Not applicable.

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
