# Peer review of "Cell-in-Cell Structures in Gastrointestinal Tumors: Biological Relevance and Clinical Applications"

_jpm, 2023, doi:10.3390/jpm13071149_

Round 1

Reviewer 1 Report

This manuscript is quite of interest and well written. Although the molecular mechanisms are still unclear, this CIC phenomenon is well described by the authors.

Comments/questions:

1)     As summarized in table-1 and -2, many types of CIC contained monocytes or macrophages in several cancers. Which is the functional significance? How these myeloid cells can be “engulfed” inside of cancer cells?

   2)     Which receptor/ligands are involved in CIC formation?

   3)     As a review, I would like to suggest to the authors to add more figures, including a figure showing the signaling pathways involved in CIC formation.

   4)     Is there any link between inflammation and CIC formation?

Overall, the manuscript is well written. Minor editing of English language is required.

Author Response

  • As summarized in table-1 and -2, many types of CIC contained monocytes or macrophages in several cancers. Which is the functional significance? How these myeloid cells can be “engulfed” inside of cancer cells?

The main functional significance of engulfment of immune cells by cancer cells is escape from the immunological control and direct killing f immune cells. The main mechanism of engulfment of immune cells is the interaction of LFA-1 receptor and ICAM-1 ligand. The corresponding explanation is in the manuscript, line from 251-260.

  • Which receptor/ligands are involved in CIC formation?

The receptor/ligands interaction for CIC formation is mainly described foe heterogenic CICs, the interaction between two cancer cells is normally accomplished through membrane integrins, such E- and P-Cadherins. Clarification is added to the manuscript. The details about different types of CIC formation are in the Section 3. Also clarifying schemes are added to the manuscript.

  • As a review, I would like to suggest to the authors to add more figures, including a figure showing the signaling pathways involved in CIC formation.

We added two more figures to the manuscripts.

   4)     Is there any link between inflammation and CIC formation?

Yes, proinflammatory environment, including different cytokines, is favorable for the CIC formation. We added corresponding clarification to the manuscript.

Reviewer 2 Report

Irina et. al. summarizes the evidence about cell-in-cell (CIC) structures with a focus on gastrointestinal tumors. It is a very intersting phenomenon of the cell-in-cell (CIC) structures. I believe that this work could be of considerable interest to the readers in the field of cell biology and cancer disease. Anyway, a major revision is required and the following questions should be addressed.
1, In figure 1, different types and signal mechanisms of CIC have been summarized. It is suggested to add some figures from literatures to support and clarify the different CIC types.
2, As the anthor said, "Numerous clinico-histopathological studies demonstrate the significance for these struc-12 tures as prognostic factor, mainly correlated with the worse prognosis."  The CIC phenomenon was observed mostly in cancer cells. Does it also often happen for the commen cells?
3, A scale bar is suggested to add in figure 2B.  It is suggest to add 2-3 figures, since a review usually contain at least 4 figures to summarize the current process.

Author Response

Comments and Suggestions for Authors

Irina et. al. summarizes the evidence about cell-in-cell (CIC) structures with a focus on gastrointestinal tumors. It is a very intersting phenomenon of the cell-in-cell (CIC) structures. I believe that this work could be of considerable interest to the readers in the field of cell biology and cancer disease. Anyway, a major revision is required and the following questions should be addressed.

1, In figure 1, different types and signal mechanisms of CIC have been summarized. It is suggested to add some figures from literatures to support and clarify the different CIC types.

We change the figure 1 to make it not so complicated and added two more figures to the manuscripts.

2, As the anthor said, "Numerous clinico-histopathological studies demonstrate the significance for these struc-12 tures as prognostic factor, mainly correlated with the worse prognosis."  The CIC phenomenon was observed mostly in cancer cells. Does it also often happen for the commen cells?

According to the literature, the frequency of occurrence of this phenomena is higher in tumor tissue, but for the common cells such phenomena could be also observed, for example in the intestinal epithelium and thymus, as well as during embryogenesis. The examples of CIC in common cells are listed in the manuscript (lines 26-40)

3, A scale bar is suggested to add in figure 2B.  It is suggest to add 2-3 figures, since a review usually contain at least 4 figures to summarize the current process.

We made the correction and added two more figures.